# The Association between Functional Dyspepsia and Metabolic Syndrome—The State of the Art

**DOI:** 10.3390/ijerph21020237

**Published:** 2024-02-18

**Authors:** Mile Volarić, Dunja Šojat, Ljiljana Trtica Majnarić, Domagoj Vučić

**Affiliations:** 1Department of Family Medicine, Faculty of Medicine, Josip Juraj Strossmayer University of Osijek, J. Huttlera 4, 31000 Osijek, Croatia; mvolaric@gmail.com (M.V.); ljiljana.majnaric@mefos.hr (L.T.M.); 2Department of Gastroenterology and Hepatology, School of Medicine, University of Mostar Clinical Hospital, University of Mostar, Bijeli Brijeg bb, 88000 Mostar, Bosnia and Herzegovina; 3Department of Cardiology, General Hospital “Dr. Josip Benčević”, A. Štampara, 35105 Slavonski Brod, Croatia; domagojvucicmedri@gmail.com

**Keywords:** gastrointestinal diseases, metabolic syndrome, gastrointestinal microbiome, stress, psychological, feeding behavior

## Abstract

Functional dyspepsia is a common functional disorder of the gastrointestinal tract that is responsible for many primary care visits. No organic changes have been found to explain its symptoms. We hypothesize that modern lifestyles and environmental factors, especially psychological stress, play a crucial role in the high prevalence of functional dyspepsia and metabolic syndrome. While gastrointestinal tract diseases are rarely linked to metabolic disorders, chronic stress, obesity-related metabolic syndrome, chronic inflammation, intestinal dysbiosis, and functional dyspepsia have significant pathophysiological associations. Functional dyspepsia, often associated with anxiety and chronic psychological stress, can activate the neuroendocrine stress axis and immune system, leading to unhealthy habits that contribute to obesity. Additionally, intestinal dysbiosis, which is commonly present in functional dyspepsia, can exacerbate systemic inflammation and obesity, further promoting metabolic syndrome-related disorders. It is worth noting that the reverse is also true: obesity-related metabolic syndrome can worsen functional dyspepsia and its associated symptoms by triggering systemic inflammation and intestinal dysbiosis, as well as negative emotions (depression) through the brain–gut axis. To understand the pathophysiology and deliver an effective treatment strategy for these two difficult-to-cure disorders, which are challenging for both caregivers and patients, a psychosocial paradigm is essential.

## 1. Introduction—Functional Gastrointestinal Disorders (Especially Functional Dyspepsia and Irritable Bowel Syndrome) and Metabolic Syndrome—Common Public Health Concerns

Functional gastrointestinal disorders (FGIDs) are the most common diagnoses in gastroenterology. More than 40% of patients worldwide suffer from FGIDs, which have an impact on healthcare utilization and quality of life. The morphological and physiological abnormalities that characterize FGIDs are discrete and include altered gut microbiota, visceral hypersensitivity, altered mucosal and immune function, and impaired central nervous system (CNS) processing [1]. As one of the most common FGIDs, functional dyspepsia (FD) has a prevalence of 20 to 40% and accounts for 3 to 5% of GP visits. FD is diagnosed when an organic cause for the symptoms cannot be found. In these patients, the gastrointestinal (GI) symptoms cannot be specifically linked to the underlying pathophysiological causes. FD is characterized by persistent or frequently recurring abdominal pain, bloating, early satiety, and epigastric burning. Over time, two-thirds of those affected suffer from persistent, irregular symptoms that can affect their quality of life and even their psychological well-being [2]. The Rome IV criteria subdivide FD into postprandial distress syndrome (PDS) and epigastric pain syndrome (EPS), with possible overlaps. Heartburn often occurs together with dyspeptic symptoms but is not the dominant problem. It is well known that FD is often associated with another functional GI disorder, such as irritable bowel syndrome (IBS) [3]. It has been found that FD patients have an approximately 8-fold increased risk of IBS compared to the general population [4].

Previous population studies have identified health behaviours such as smoking, risky alcohol consumption, inadequate physical activity, poor diet, anxiety and depression, experiencing stressful situations, and increased sensitivity to stress as factors in the development of FD [4]. In particular, patients with FD often report higher levels of psychological stress compared to healthy controls, and higher levels of stress are associated with higher symptom severity, suggesting a direct link between stress and dyspeptic symptoms [5]. Although psychological comorbidity is common, it is not yet clear whether it precedes or is influenced by FD symptoms. 

Helicobacter pylori (HP) infection and long-term use of nonsteroidal anti-inflammatory drugs are considered the most important factors in the development of dyspepsia caused by organic GI diseases, gastritis, and ulcer disease, but psychological stress and lower socioeconomic status, including a lower level of education, are also known to play an important role in the development of these diseases. Their development is assumed to be the effect of poor health habits (smoking, excessive alcohol consumption, insufficient physical activity, unhealthy eating habits), as well as psychological disorders such as depression and anxiety that frequently accompany a poor socioeconomic background. Some personality traits show strong correlation with these diseases, such as introversion, neuroticism, psychoticism, and hostility, in addition to altered sensitivity to stressful situations. The aforementioned psychosocial factors greatly affect the biological mechanisms of the disease, such as by activating the HP infection, but also by further impairing psychological well-being, thus enabling a greater prevalence of unhealthy habits and resulting in an increase in the risk of disease [3]. 

This understanding is similar to that of how FGIDs originated. As our understanding grows of the pathophysiology of the emergence of FGIDs, such as FD, it becomes clear that many of the aforementioned factors can be precisely modified by psychosocial influences, thanks to the numerous connections between the brain and the GI system. Studies have provided an overview of how the complex interactions of environmental, psychological, and biological factors contribute to the development and maintenance of FGIDs, thereby affirming the integration of biopsychosocial factors in the emergence of symptoms, as well as an individualized approach to the treatment of the said disorders due to their complexity of origin [6,7].

Early childhood experiences, trauma, illness, social stress, and lack of support are genetic and environmental variables that have an impact on the brain and the gut. They frequently generate anxiety and depression, followed by somatization, but they can also cause structural changes in the CNS, such as modification of visceral afferent impulses that are critical for emotions and cognition. The direct influence on intestinal physiology suggests that unhealthy behaviors affect motility, permeability, inflammation, and bacterial flora, but this effect can also manifest indirectly via the brain–gut axis. As a result, the ANS and the HPA axis enhance bidirectional communication between the brain and the gut. The gut bacteria communicate with the brain in both directions via neurological, endocrine, and immunological pathways; this communication is essential for anxiety, depression, behavioral, and cognitive disorders, as well as persistent visceral discomfort. The combined effects of changed physiology and the individual’s psychological condition will define the sickness experience, the disorder’s severity, and, eventually, the clinical result. Psychosocial aspects are critical for understanding FD pathogenesis and developing a successful treatment approach and knowledge of this biopsychosocial paradigm, and are especially essential for assessing and treating difficult-to-treat illnesses, which frequently cause inconsistencies and discomfort in caregivers and patients [8]. 

It appears that the immune system of the mucosa of the GI tract, which is characterized by infiltration with eosinophils and mast cells, plays an important role in this link between psychological stress and the symptoms of FD [9]. Eosinophils and mast cells are thought to link innate and adaptive immunity to maintain homeostasis of the intestinal epithelium [7]. Under normal conditions, the entire GI tract, except the healthy esophagus, is colonized with eosinophils. While eosinophils are most commonly involved in allergic diseases such as asthma, rhinitis, and atopic dermatitis, their increased infiltration can also occur in certain GI disorders such as eosinophilic GI disorders, inflammatory bowel disease (IBD), and gut–brain interaction disorders, often colocalizing with mast cells in both homeostatic and inflammatory conditions (eosinophil–mast cell axis) [10,11,12,13]. Once activated, eosinophils release granules consisting of major basic protein (MBP), eosinophil peroxidase (EPO), eosinophil cationic protein (ECP), and eosinophil-derived neurotoxin (EDN), which enhance the release of pro-inflammatory cytokines, leading to changes in smooth muscle contraction, increased vascular permeability, visceral hypersensitivity, and promotion of Th1-Th2 polarization [14,15]. Recent studies in animal models clearly show that psychological stress exacerbates intestinal inflammation by triggering the degranulation of eosinophils and mast cells [16]. Increased eosinophil infiltration has also been found in the duodenal mucosa of patients with FD [9]. Eosinophil infiltration is estimated to cause low-grade inflammation in up to 40% of FD patients, and when these cells degranulate, symptoms occur along with impaired mucosal integrity and structural and neuronal abnormalities [17,18]. These findings suggest an involvement of the immune system in FD. This is also confirmed by a decrease in soluble factors that are indicators of duodenal mucosal barrier function, such as zonula occludens-1 (a major tight junction protein) and claudin 2 and 4 (contribute to impaired duodenal mucosal barrier function) [19,20]. In addition, an increase in inducible nitric oxide synthase (iNOs), a result of mast cell degranulation, is associated with the sensation of bloating in the upper abdomen, while the increase in the neurotrophic factor neurotrophin is associated with the sensation of burning in the epigastrium [21,22].

Previous studies on the relationship between chronic psychological stress and somatic diseases have shown that stress-induced inflammation and oxidative stress are the main factors in the body’s pathophysiological response to chronic stress [23]. To clarify this complex relationship, it is necessary to understand the neurobiology of psychological stress, which is based on the mutual interaction of cortical-limbic brain structures responsible for the control of emotions and behavior, the hypothalamic–pituitary–adrenal (HPA) axis, and the autonomic nervous system (ANS). Frequent or long-term activation of the HPA axis leads to excessive secretion of cortisol and increased sympathetic nervous system activity, resulting in a range of pathological conditions [24].

Chronic psychological stress and other aspects of modern lifestyles, including unhealthy diets and widespread use of antibiotics, are increasingly recognized as a common background to the associations of the epidemic of obesity and its associated diseases, metabolic syndrome (MetS), type 2 diabetes (T2D), and cardiovascular disease (CVD) on the one hand, and FGIDs on the other, through mechanisms such as an imbalance in the gut microbiome and increased systemic inflammation [25]. The neurobiological background was found in the existence of the brain–gut axis [26]. This is a bidirectional communication network between the gut and the CNS that includes nervous, hormonal, metabolic, and immune-mediated mechanisms (Figure 1).

The theoretical link between the CNS and the immune system of the GI tract is supported by the presence and function of glial cells in the gut [27]. Due to their secretory ability, influenced by inflammatory mediators, intestinal glial cells serve as a link between the nervous and immune systems. In addition, the stress-induced HPA axis leads to the release of various stress-related mediators such as corticotrophin releasing factor (CRF), which reduces the number of CD4 T lymphocytes and the expression of tight junction proteins in the duodenal mucosa; then, glucocorticoids, mineralocorticoids, serotonin, glutamate, and gamma-aminobutyric acid, which affect the concentration of immune cells and the expression of tight junction proteins in the duodenal mucosa [28]. Glial cells themselves have the ability to present antigens, control intestinal smooth muscle function, and secrete pro-inflammatory cytokines such as tumor necrosis factor-alpha (TNF-α), interleukin (IL)-1 and -6, and glial cell-line-derived neurotrophic factor [29,30]. It is assumed that the glial cells of the intestine release neurotrophic factor as an inflammation modulator in response to inflammation in the duodenum and thereby protect the glial cells [22]. In addition, CRF can be synthesized not only in hypothalamic cells but also in eosinophils under the influence of stress [31].

Our hypothesis is therefore that environmental factors associated with modern lifestyles, including in particular increased levels of psychological stress, may explain the significant increase in the incidence of MetS and FGIDs, especially FD, in recent decades. This link is supported by the growing body of evidence on the pathophysiological links between chronic stress, obesity-related MetS, gut dysbiosis, increased inflammation, and FGIDs, implying that these two conditions may be mutually reinforcing under conditions of chronic stress.

## 2. The Potential Influence of the GI System on the Development of Obesity/MetS

Commensal bacteria colonize the free surfaces of the host, especially the lumen of the intestine, mainly the colon, where commensal bacteria are most densely populated (about 1000 per 1 cm of intestinal content). The microbial community is diverse, and dominated by two large families of bacteria: Firmicutes and Bacteroidetes. These are also Lactobacilli, Mollicutes, and Proteobacteria. Colonization of the gut with commensal bacteria begins during childbirth. The microbiome population stabilizes between the ages of two and three and stays unchanged until adulthood. The way of giving birth and the early postnatal course of life greatly influence the development of the intestinal microflora, so cesarean delivery, premature birth, infections, and taking antibiotics in early childhood significantly impair microflora development. The child’s mucosal and systemic immune system is developing concurrently and in conjunction with the commensal microflora. Large amounts of antigenic material found in commensal intestinal bacteria interact with nonspecific (innate) immune system receptors to stimulate the growth of intestinal lymphatic tissue. In this process, intestinal bacteria create a type of “mold” that supports the immune system’s continued development [32].

The intestinal microbiome forms an interactive homeostatic ecosystem with the host and influences human health. In addition to the “healthy” gut microbiome, which is an important element in maintaining host homeostasis, an imbalance of the gut microbiota has been associated with the onset and progression of symptoms in patients with FD and IBS [33,34]. In this sense, the gut microbiome can be divided into the luminal microbiota and the mucosa-associated microbiota, which represent two functionally and taxonomically distinct subunits, with the mucosal component playing an important role in the pathogenesis of GI disorders due to its proximity to the epithelium and acting as a barrier that prevents the passage of pathogens [33,35]. By allowing parts of the intestinal microbiome to seep through gaps in the gut wall, the microbiome modifies the mucosal immune system. The microflora also has an indirect influence on systemic immunity [32,36,37]. In this context, there is a general difference in the bacterial composition of gastric contents between patients with FD and healthy individuals, suggesting that the reflux of intestinal contents and gut bacteria into the stomach may be the pathophysiologic basis of FD [38]. Although this view may explain the symptoms of FD, it does not identify the cause of reflux, which poses a challenge to the full understanding of the disease and its exact pathophysiologic mechanism.

The microbiota of the human gut varies according to factors such as age, sex, ethnicity, method of birth, and neurobiological processes. Environmental factors such as infections, an unhealthy lifestyle with a low-fiber diet, and drastic dietary changes are also important. The effects of antibiotic use are particularly important as they can alter the diversity and composition of a person’s microbiome, which can have direct and long-term effects on health [36,39,40,41,42]. Studies suggest that the pathophysiology of many diseases, including MetS and GI diseases, is highly dependent on the diversity of the intestinal microbiota and the relationship between the human immune system and the microbiota. The intestinal microbiota can influence physiological functions such as gut motility, intestinal barrier permeability, nutrient absorption, hormone and/or neurotransmitter production, and fat storage. There are differences in the composition of the gut microbiota between healthy people and people with metabolic, immunological, and neurological diseases [39,40,43,44].

The MetS is defined as a group of metabolic dysregulations, frequently matched with central obesity, insulin resistance (IR), atherogenic dyslipidemia, and hypertension, that are strongly connected to an increased risk of T2D and CVD if not treated [45]. Although the etiology and pathophysiology of MetS are the result of a complex interplay of dietary habits, lifestyle, environmental factors, and genetics, the role of dysbiosis of the intestinal microbiota is undoubtedly important, and more and more studies are focusing on discovering the mechanisms of this influence. In this sense, studies have shown that people with MetS have a different microbiome than healthy people due to the excessive growth in pathogenic bacteria and the suppression of the growth in beneficial bacteria. The reason for this is that injury to the intestinal barrier can lead to intestinal inflammation and affect the interaction between the gut microbiota and the host [46]. Studies suggest that there is a link between the composition and diversity of the gut microbiota and systemic low-grade inflammation, which contributes to the development of obesity and MetS by sending proinflammatory signals to the host [47,48]. Lipopolysaccharides (LPSs), also called endotoxins, which are derived from the outer membrane of Gram-negative bacteria, are thought to be the first step in the inflammatory processes associated with the development of obesity and MetS. This is because lipid A, a structural component of LPS, can penetrate the chylomicrons or leaky intestinal tight junctions to pass through the GI mucosa. LPS invades tissues such as the liver and adipose tissue after entering the bloodstream, which triggers an innate immune response [49]. When LPS binds to the LPS-binding plasma protein (LBP), it activates the receptor protein CD14 on the plasma membrane of macrophages. The resulting complex binds to the surface innate Toll-like receptor 4 (TLR4), which triggers intracellular transduction signals that drive the expression of genes encoding multiple inflammatory effectors, ultimately leading to activation of the inflammasome and multiprotein oligomers that collect inflammatory signals, and triggering the innate immune system [50].

There is a complex reciprocal relationship between obesity/MetS and the intestinal microbiome/FGIDs (Figure 2). Unhealthy dietary habits and nutrient intermediates resulting from overfeeding can reduce the diversity of the gut microbiota. For example, a high-fiber diet leads to an increase in Bacterioidetes, while in obese people, on the contrary, an increased proportion of Firmicutes has been observed, which is associated with a higher calorie intake from food and further weight gain [51,52]. Studies on leptin-deficient mice show a dysbiosis of the intestinal microbiome that is comparable to obesity caused by a high-fat diet. Complementarily, recipients of transplanted microbiomes from obese mice with leptin deficiency also become obese [51].

The role of the microbiota–gut–brain axis also should not be neglected. This implies a complex communication between the gut and the brain via molecules and metabolites that regulate both CNS and gut functions [53]. There are also wired communication channels via the ANS and the enteric nervous system (ENS) (Figure 1). Through this network, the microbiota can communicate with the brain and influence the development of various diseases, behavior, mental health, and even an individual’s preferred food choices [53,54]. The intestinal microbiota can play a regulatory role by directly or indirectly influencing the production of neuroactive metabolites [55,56]. Studies in mice have shown that when the microbiome of humans with depression and anxiety is transplanted, symptoms appear in the recipient mice [57]. Some studies conducted on humans with anxiety and depression show a positive effect of the use of probiotics and prebiotics on the disappearance of anxiety and depression symptoms, and some studies even suggest a possible implementation as an add-on therapy to the currently available pharmacological therapy [58,59,60].

The potential therapeutic method that is the subject of much debate is fecal microbiota transplantation (FMT). This is a method in which bacteria from the intestines of healthy people are transferred to sick patients [61]. It currently plays a role in the treatment of recurrent and refractory Clostridium difficile infections, but many research results demonstrate its high potential in curing IBD and some autoimmune diseases, or in correcting the dysbiosis of the gut microbiome after antibiotic treatment [62,63]. Both FMT and the use of prebiotics and probiotics are promising methods in human pathology, but still require further research to fully understand the mechanisms of impact on human health and various diseases, as well as the reduction in unwanted complications. However, these therapies will likely also play a role in the treatment of obesity and MetS in the future.

## 3. The Influence of Obesity, Especially Visceral Abdominal Obesity, and MetS on Increased Degree of Systemic Inflammation and the Development of Functional Dyspepsia

A subset of obese people known as “metabolically healthy obese” constitutes about 20% of the total obese population. In comparison to the “at-risk” obese group, with high visceral obesity, they have reduced visceral fat content, higher insulin sensitivity, and a favorable metabolic profile. Many studies outline the data for excess visceral adiposity as a predictor of IR and a proatherogenic, pro-inflammatory profile associated with many other diseases [64,65,66].

Visceral obesity is characterized by chronic inflammation of the adipose tissue. Studies have shown that the expansion of adipose tissue macrophages (ATMs) is a major contributor to IR and metabolic dysfunction in obese people. ATMs are immune cells that secrete pro-inflammatory cytokines, galectin-3, and exosomes. Normally, most resident ATMs have an M2-like polarized phenotype which encourages tissue remodeling and healing. Regulatory T (Treg) cells and eosinophil-released factors help ATMs sustain this anti-inflammatory state. However, in obese individuals, the proportion of M1-like polarized ATMs, responsible for inflammation, is significantly higher, shifting the balance towards a tissue pro-inflammatory state. This is due to the higher number of M1- and M2-like polarized ATMs in obese individuals [67,68]. Increased adipocyte chemokine production leads to the recruitment of blood monocytes and ATM proliferation. Monocyte-derived ATMs mainly have the M1-like polarized phenotype expressed by CD11c or CD9. Inflammation and M1-like ATM polarization are enhanced by the increases in neutrophils, innate lymphoid cell type 1 (ILC1) cells, CD8+ T cells, Th1 cells, and B2 cells, and the decrease in eosinophils and Treg cells [68,69].

It is widely accepted that obesity-induced pro-inflammatory ATMs contribute to IR. However, the specific mechanism by which M1-like ATMs affect reduced insulin sensitivity is still unclear, even though various studies have been conducted. It is reasonable to assume that pro-inflammatory ATMs produce substances such as TNF-α, IL-6, galectin-3, and others, which may have paracrine or systemic effects on insulin target cells, compromising insulin signaling. Recent research has shown that exosomes released by ATMs also play a significant role in the development of IR. These exosomes contain miRNA (small non-coding RNA molecules that regulate gene expression) that control the body’s sensitivity to insulin. Injecting exosomes produced from lean mouse ATMs has increased insulin sensitivity in obese mice, while exosomes derived from obese mouse ATMs have induced IR in lean mice. Additionally, applying “lean” or “obese” exosomes directly to insulin target cells in vitro has resulted in insulin sensitivity or IR, respectively. However, this therapy still needs confirmation in humans [67,68].

Adipose tissue associated with obesity contains various types of immune cells in addition to secretory active macrophages [70]. The adipose tissue of obese individuals has been found to have an increase in pro-inflammatory CD8+ and CD4+ Th1 cells, and a decrease in anti-inflammatory CD4+ Th2 lymphocytes, compared to non-obese individuals. Recent research suggests that there is a shift in balance from anti-inflammatory CD4+ Treg cells towards pro-inflammatory CD4+ Th17 lymphocytes in obese individuals, which are involved in obesity-related pathologies and associated with a high degree of IR. Leptin, adiponectin, angiotensinogen, plasminogen activator inhibitor (PAI)-1, IL-6, and TNF-α are among the recognized mediators responsible for the pathophysiology of IR and MetS [70,71].

Different diets may differently regulate immuno-metabolic pathways in adipose tissue. While overfeeding and a Western diet are associated with fat storage in the visceral adipose tissue and systemic inflammation (driven by Th1/Th17 immune response), physical activity and a Mediterranean diet, along with the availability of essential micronutrients, skew the adipose tissue immune response toward type 2 inflammation (Th2 cell dependent), which results in a decrease in this adipose tissue inflammation, thus preventing weight gain [72].

MetS is characterized by higher levels of inflammation compared to general obesity. The pathogenic processes that lead to the development of MetS result in a pro-inflammatory state. This explains why people with MetS have higher levels of inflammatory markers such as TNF-α, C-reactive protein (CRP), and IL-6. Studies have shown that obesity and IR increase IL-6 levels [73,74]. It is well-known that IL-6 controls the metabolism of fat and glucose, influencing IR through intricate processes. Production of TNF-α is also linked to IR. TNF-α promotes hepatic lipolysis, which raises free fatty acid (FFA) levels and affects insulin signaling in adipocytes and hepatocytes through serine phosphorylation and insulin receptor deactivation [75]. Higher levels of inflammatory mediators are associated with increased body fat tissue. The high levels of pro-inflammatory cytokines in patients with a higher body mass index (BMI) could explain why older people with MetS are more likely to be obese than those without MetS [76]. Researchers are focusing on the basic pathophysiology of adipose tissue malfunction to discover novel targets for the therapy of MetS. The complement system and the inflammasome are two key areas of interest. Early adipose tissue inflammation is primarily regulated by the inflammasome, which acts as the second messenger by converting the cell’s stress signals into innate immune responses. The resulting increased production of pro-inflammatory cytokines causes IR and MetS, exacerbating metabolic disturbances and raising the level of inflammation even further [77,78].

Vascular changes due to endothelial dysfunction, which is common in MetS patients with high blood pressure, can contribute to the development of metabolic disturbances and inflammation. This is because of microcirculation dysfunction, which impairs insulin supply to target tissues and endothelial-related inflammation. Obesity promotes endothelial dysfunction through various metabolic factors and shear-stress forces caused by the accumulation of adipose tissue, high blood pressure, dyslipidemia, and diabetes. These factors lead to increased vascular oxidative stress and a reduction in the availability of nitric oxide (NO), which is the main vasodilator factor. Studies have discovered a significant link between pro-inflammatory cytokines present in MetS and visceral obesity, and endothelial dysfunction. After six weeks of a high-fat diet, pro-inflammatory cytokine levels began to rise, whereas adiponectin levels fell. Endothelial function and serum NO also declined after six weeks of a high-fat diet, and total visceral fat mass was adversely connected to endothelial function [79,80]. In precision medicine, pathophysiological mechanisms involved in the pathogenesis of obesity-related hypertension, such as leptin resistance, impaired baroreceptor and chemoreceptor reflexes, increased renal sympathetic nervous activity, mitochondrial dysfunction, and the regulatory role of intermedin and adrenomedullin, can be the focus of specific and selective therapeutic interventions [81].

Research suggests that dietary patterns can affect systemic inflammation by affecting metabolic pathways either directly or indirectly, via influence on the gut microbiome composition and metabolite production. Studies have found that consuming a Western diet, which includes high amounts of red meat and sweets, is associated with a higher body mass index (BMI) and fat mass. On the other hand, a healthy diet with a lower intake of saturated fatty acids (SFAs) and trans fatty acids (TFAs) has been shown to decrease pro-inflammatory indicators like serum LPS, and is inversely related to IR [82]. Metabolic inflammation, also known as metainflammation, is a low-grade systemic inflammation that is often seen in metabolic disorders related to obesity. It is believed that various harmful dietary factors, including SFAs and certain carbohydrates, can trigger metainflammation. The type of diet can also influence immune regulation indirectly, by affecting the microbiome composition, which in turn releases a wide range of microbial metabolites, with profound effects on neural and immune regulation [83]. This implies that certain dietary factors may be more significant in causing inflammation than just overeating and resulting in obesity [84].

There is increasing evidence that links obesity, particularly abdominal visceral obesity, to a higher risk of developing GI diseases such as gastroesophageal reflux disease (GERD), erosive esophagitis, non-alcoholic fatty liver disease (NAFLD), FD, and GI cancers. Obesity not only increases the chances of getting GI disorders, but it also leads to more severe forms of the disease and a poorer response to treatment, leading to more negative clinical outcomes and increased clinical and economic burden [85]. Obesity and GI disease may be linked through various pathways, including mechanical, nutritional, inflammatory, and pro-cancerogenic variables.

Obesity causes IR through increased FFAs, pro-inflammatory cytokines, and altered adipokines, which can result in metabolic changes and increased systemic inflammation, known to play a role in both benign and malignant GI disorders. Obesity’s mechanical effects may also contribute to esophageal disease and other GI symptoms, while a recent study found that adipocyte-secreted peptides such as leptin, adiponectin, nesfatin-1, and apelin can affect GI motility by acting both centrally and peripherally. The knowledge is growing on the function of mechanosensitive ion channels in the GI tract, which transfer mechanical stimuli into biochemical signals, to coordinate peristaltic contractions with food digestion and nutrient absorption. Mechanosensation is essential for normal GI tract function, while abnormalities are associated with GI disorders [86].

Obesity is associated with both quantitative and qualitative changes in the gut microbiota of obese patients, which in turn determine the activation of various pathophysiological pathways. Thus, an imbalance in the two most abundant bacterial species in the intestines, Bacteroidetes and Firmicutes, with a higher proportion of Bacteroidetes, leads to increased methane production. This, in turn, causes an increase in intraluminal pressure in the intestines, which increases the risk of developing diverticulosis. Furthermore, changes in bacterial representation cause disruptions in lactose fermentation and digestibility, leading to IBS. Increased breakdown of complex carbohydrates allows for greater calorie intake from food, promoting obesity [85].

Obesity-related inflammation can increase the permeability of the gut, causing the tight junctions of the intestine to become leaky. This can allow bacteria and their components to pass through the intestinal barrier and affect the metabolism of mesenteric adipose tissue in various ways. The production of LPS by microbiota causes an inflammatory response in adipocytes, which enhances the recruitment of ATMs in the mesenteric fat depot. LPS also interacts with TLR-4 to promote the transcription of pro-inflammatory genes in ATMs [85]. The innate immune and inflammatory cells in the mesenteric fat are constantly stimulated, leading to the activation of lymphoid cells. In cases of FD, this is mainly manifested by eosinophil and mastocyte infiltrates in the duodenum. Bacterial stimuli may cause local activation of the peroxisome proliferator-activated receptor (PPAR)-γ, inducing adipocyte proliferation and differentiation in the mesenteric fat depot. These factors all act together, aggravating abdominal obesity [87].

When the barrier function of the gut is compromised, LPS (endotoxin) may enter the systemic circulation, as it surpasses the capacity of Kupffer cells in the liver. LPS binds to LBP, which then binds to CD14, mostly expressed by macrophages, to activate an immunological response via the TLR4 and intracellular transcription factor of the nuclear factor-B pathways. This leads to an increase in circulating LPS levels, resulting in metabolic endotoxemia, characterized by a low-grade pro-inflammatory condition [88].

Obesity is associated with an increased risk of anxiety/depression. The gut microbiome helps regulate the gut–brain axis and maintain health, while dysbiosis is linked to obesity and has negative effects on mood and cognition. Multiple pathways of the gut–brain axis, including metabolic, immune, hormonal, and neural signals, make a connection between obesity and anxiety/depression (Figure 1) [26]. In addition, systemic inflammation may lead to the breakdown of the blood–brain barrier, allowing different metabolites and immune system components to enter the CNS and promote neuroinflammation and neural pathway alterations [89]. The CNS alterations such as dysregulation of the HPA axis, with changes in glucocorticoid production and in levels of neurotransmitters, leading to activation of a pro-inflammatory milieu and changes in behavior, are responsible for the progression of obesity and eventually may cause functional GI disorders [88] (Figure 1 and Figure 3).

## 4. Chronic Psychological Stress—A Common Denominator of MetS and FD

Persistent psychological stress can lead to the development of unhealthy habits, which can ultimately cause IR, abdominal obesity, and dysbiosis of the intestinal microbiota. These unhealthy habits can include consuming foods that are high in fat and carbohydrates, as well as relying on coping mechanisms like alcohol consumption, smoking, or using psychotropic drugs to reduce stress-related anxiety and tension. Such behaviors can influence health by causing pathophysiological changes. These harmful behaviors are a result of the impact of long-term psychological stress on the cortical-limbic regions of the brain [90,91] (Figure 3). This stress raises glucocorticoid levels, increases sensitivity to nicotine, modifies the CRF stress response, encourages ethanol use, and increases the number and occupancy of glucocorticoid receptors [92]. All these mechanisms may explain the connection between persistent stress and a lack of physical resilience, and the role that unhealthy behaviors play in linking them [23].

There are three main ways in which the body responds to stress: the renin–angiotensin system (RAS), the ANS, and the HPA axis. Studies show that experiencing stressful life events or long-term psychological stress can increase the risk of IR and MetS/T2D. Many different factors contribute to this link, including ANS, RAS, lipid metabolism, immune response, pancreatic beta cells, and stress-related hormones [24].

Prolonged psychological stress can affect the functioning of beta cells, which are responsible for regulating glucose levels in the body. This can occur when genes that control beta cell proliferation are suppressed, leading to a reduction in the functional mass of pancreatic islet cells and ultimately causing islet atrophy [93]. Chronic stress can also affect lipid metabolism by hyperactivating the ANS, disrupting normal glucose homeostasis and insulin signal transduction. This can also result in dysregulation of glucose metabolism. Additionally, prolonged stress can lead to overstimulation of the immune system, which can increase levels of inflammatory mediators and recruit pro-inflammatory cytokines and/or pro-apoptotic agents into the pancreatic islets. Endocrine disorders, such as excessive cortisol and low sex steroid levels, which are common in chronic stress, contribute to insulin’s disability to exert its hyperglycemic effect [93,94].

Subclinical hypercortisolism may provide a biological explanation for the link between continuous stress and MetS/T2D. High cortisol levels increase visceral adiposity by promoting adipocyte differentiation and proliferation. This leads to fat redistribution from peripheral to central depots, increases in the size and number of adipocytes, lipolysis, release of FFAs, and glucometabolic disturbances. These effects are caused by the glucocorticoid receptors, which are more prominent in visceral adipose tissue than subcutaneous adipose tissue [95]. Cortisol is a potent inducer of IR in all insulin-sensitive organs. It alters the primary anabolic insulin pathway by using multiple molecular pathways from the insulin receptor to transcription factors. This insulin-impaired response leads to reduced glucose absorption by muscle and adipose tissues. If the insulin-resistant state persists for months or years, the beta cells work hard to create extra insulin to counteract the glucose overload, which leads to their exhaustion [94].

Abdominal visceral obesity caused by unhealthy eating habits and stress is often linked to IR. This type of obesity attracts immune cells like macrophages, which intensify systemic inflammation and lead to further metabolic derangement. When dead hypertrophic adipocytes and hypoxia are present, macrophages are drawn to this area to repair the damage. Most of the macrophages recruited in this process form crown structures. Due to their smaller size, a large quantity of lipids enters macrophages and slows down the process of exocytosis. This could be the reason why pro-inflammatory M1 macrophages polarize from anti-inflammatory M2 macrophages. Dysfunctional adipocytes induce a variety of inflammatory mediators such as leptin, TNF-α, IL-6, IL-1β, and FFAs [96]. Long-term psychological stress can lead to obesity, IR, and increased inflammation, which in turn can cause MetS or T2D. This is because of its behavioral effects, which include changing eating habits and engaging in other unhealthy behaviors, as well as its effects on the HPA axis and the ANS.

Within this context, we can consider the development, recurrence, and persistence of FD symptoms. These symptoms depend on complex circuits involving the interaction of the gut microbial contents, epithelial cell permeability, gut wall immune cell infiltration, mediators released from the gut nerves, and systemic stress-related mechanisms [97]. Unhealthy eating habits and other unhealthy behaviors caused by long-term psychological stress have an impact on the development of gut dysbiosis and the impairment of the GI system’s motility and epithelium permeability, which results in increased inflammation, leading to abdominal obesity, MetS, and illnesses like FD. Impaired duodenal barrier integrity and the intensity of the gut wall immune cell infiltration are correlated. The development and expression of FD symptoms are significantly influenced by complex bidirectional gut–brain communications; psychological stress affects both the gut and the brain [98] (Figure 1 and Figure 3).

Psychological stress increases gut permeability by activating mast cells through the action of CRF. The study showed that in response to stressors, intestinal paracellular permeability rises, causing visceral hypersensitivity and bacterial translocation. The inflammatory response might be triggered by an increase in intestinal permeability, even before mucosal inflammation becomes measurable, leading to a feed-forward cycle between the dysfunctional epithelial barrier and inflammatory responses. This could then keep the low-grade inflammatory response going and make it worse [99].

There is a connection between increased gut permeability and leptin resistance, which is mediated by chronic inflammation. This connection promotes both obesity and IR, leading to an association of MetS with FD [100]. Chronic inflammation caused by obesity, and unhealthy eating habits, can disrupt enterocyte secretion, leading to abnormalities in tight junctions. This, in turn, causes an increase in the leakage of bacteria and their byproducts across the intestinal mucosa. The numbers of bacterial endotoxins in the blood can also increase, aggravating systemic inflammation and worsening IR, which is closely linked to chronic inflammation [101].

As we gain knowledge about the complexity of the gut microbiota, it has become evident that immunological and metabolic responses, as well as microbe–host interaction, contribute significantly to GI motility disorders. The bacteria in the intestinal microbiome produce SCFAs that activate the enzyme tryptophan hydroxylase 1 (TPH1), which leads to the production and secretion of 5-hydroxytryptamine (5-HT) from intestinal enterochromaffin cells (ECCs). The basal membrane of intestinal ECCs releases 5-HT, which interacts with receptors on neurons in the ENS, modulating intestinal motility [102].

It is possible to argue that chronic stress is the root cause of disorders such as MetS/T2D and FD. This stress modifies behavior and affects various systems in the body, including the HPA axis, ANS, RAS, and ENS through the gut–brain axis (Figure 1). This leads to pathophysiological changes that are common to both MetS/T2D and FD, such as obesity, IR, changes in the microbiome, gut motility, and permeability of the intestinal epithelium, and increased levels of inflammation. Additionally, both conditions can further exacerbate each other through the mechanism of increased inflammation (Figure 3).

## 5. A Brief Overview

This review of the state of the art provides insights into the immune and metabolic responses, as well as microbe–host interactions, that play a significant role in FGIDs. Among these, FD has become more prevalent in recent years. Although GI diseases are usually not related to metabolic disorders, they are connected by several pathophysiological pathways. Chronic psychological stress, unhealthy lifestyles, and antibiotic use have all been associated with obesity and related conditions such as MetS, T2D, CVD, and FGIDs. These concerns are influenced by an imbalance in gut microbiota, increased inflammation, and the brain–gut axis, which connects the intestine to the CNS. These factors contribute to the obesity epidemic and associated illnesses.

The presence of MetS and IR causes chronic low-grade inflammation throughout the body. This also leads to changes in the microbiome of the intestine, which worsen the GI motility disorder. This causes functional problems in the GI system and allows for further worsening of chronic systemic inflammation by increasing the permeability of the intestinal mucosa for Gram-negative bacteria.

The gut microbiota helps to control the gut–brain axis and maintain health, while dysbiosis is connected to obesity and its negative effects on mood and cognition due to poor lifestyle choices and stress. Western food habits appear to limit microbial diversity, promote inflammation, and lead to leaky gut syndrome, which can cause peripheral inflammation and changes in the CNS.

Chronic psychological stress may be the common cause of both MetS and FD. This is because it modifies behavior and acts on the HPA axis, ANS, RAS, and ENS via the gut–brain axis. This leads to pathophysiological changes that are common to both conditions, such as obesity, IR, changes in the microbiome, gut motility, and permeability of the intestinal epithelium, and increased levels of inflammation. Furthermore, both illnesses can exacerbate each other via the mechanism of increased inflammation.

FD itself is often associated with anxiety and chronic stress, which can activate the neuroendocrine stress axis and, thus, the immune system. This promotes unhealthy habits that contribute to the development of MetS and disruption of the intestinal microbiome. Based on this, we support the concept of an integrated model of MetS and FD to enable a comprehensive care program and pave the way for innovative therapeutic options.

## 6. Future Perspectives

With this work, we hoped to raise awareness among researchers and the general public about the existence of common risk factors and pathophysiological pathways for FD and MetS, as well as to encourage, based on all of the above, consideration of using innovative treatment methods that will be able to efficiently cure both disorders.

These methods include non-pharmacological forms of treatment, such as adopting the Mediterranean diet or implementing complementary forms of treatment from Eastern traditional medicine, such as traditional Chinese herbal therapy or acupuncture treatment, which have anti-inflammatory effects and numerous effects on intestinal motility [103,104,105,106]. For example, it is known that phytochemicals contained in cruciferous vegetables include an important amino acid tryptophan, which has several immune-modulatory effects [107].

Avoiding particular foods may also be useful. Studies have revealed that a high-fat meal causes more nausea and discomfort in FD patients than a high-carbohydrate or control meal, indicating that reducing dietary fat may be useful for FD therapy [108]. It has also been proposed that emulsifiers, which are added to most processed foods to enhance texture and lengthen shelf life, may have contributed to the significant rise in the occurrence of chronic inflammatory illnesses, such as FD and MetS [109]. Studies have revealed that probiotics may be a helpful therapeutic approach for patients with FD [110]. The administration of probiotics lowered the severity of symptoms such as stomach discomfort and bloating. The consumption of probiotics was linked to an increase in good metabolites such as pelargonic acid, benzoic acid, and SCFAs, whereas detrimental gut metabolites like hippuric acid declined [111]. In the upper GI tract, probiotics act by lowering the levels of Escherichia/Shigella, a major source of toxic LPS, which results in the restoration of the changed gastric microbiome. This further appears to reduce visceral hypersensitivity by regulating pain receptor expression in the GI tract, which has an impact on gut motility [112].

It has long been known that ion channels regulate energy homeostasis and the progression of metabolic disorders, e.g., MetS, and today’s therapy for many metabolic diseases is based precisely on the treatment of disorders in the function of the mentioned ion channels [113]. Ion channels also play an important role in visceral hypersensitivity, motility, and intestinal permeability changes associated with FGIDs. Genetic mutations and aberrant functional expression of ion channel subunits can result in channelopathies. These channelopathies in gastroenterology are gaining popularity, and evidence of co-relationships with metabolic disorders is growing. Various findings are available. Mutations in the ABCC7/CFTR gene, for example, have been associated with constipation and diarrhea. Mutations in the SCN5A gene are instead linked to IBS. Mutations in the TRPV1 and TRPA genes of the transient receptor potential subfamily cause hypersensitivity and visceral discomfort in sensory neurons. The identification of a link between channelopathies and FGIDs offers new pathways for identifying novel direct therapeutic targets for particular channelopathies, with important implications for diagnosing and treating FGIDs [86,114].

Cognitive behavioral therapy, motivational interviewing, and mindfulness-based strategies can also assist people with metabolic illnesses and FD to achieve long-term lifestyle adjustments. These therapies aim to address psychological obstacles, increase self-efficacy, reduce stress, and facilitate behavior transformation to enhance long-term adherence to good behaviors, which can contribute to FD and MetS treatment [115].

## 7. Conclusions

FD, one of the most frequent FGIDs, and MetS, whose incidence has risen in recent decades, have many pathophysiological similarities. Environmental factors stand out, particularly those associated with the contemporary lifestyle, including poor eating habits, a lack of physical exercise, and an increase in psychological stress. The illnesses indicated above interact with one another, causing both ailments to deteriorate. Systemic chronic inflammation and disruption of the intestinal microbiome in MetS cause or support the development of FGIDs. An indispensable factor is the effect of psychological disorders such as anxiety, depression, and negative emotions, often present in MetS, on the brain–gut axis, which further aggravates the symptoms of functional disorders of the GI system. Similarly, FGIDs are frequently associated with increased psychological stress and an individual’s altered emotional response to stressful situations, which leads to activation of the gut–brain axis, changes in the immune system, and changes in the gut microbiome, resulting in further deterioration in the development of unhealthy habits and contributing to MetS. The above serves as the foundation for accepting a common biopsychosocial model of the origins of these disorders, allowing the further development of therapeutic options that, in addition to the therapy used thus far, will seek to influence all segments of complex pathophysiological causes, including psychological therapy and the treatment of intestinal microbiome disorders.

## Figures and Tables

**Figure 1 ijerph-21-00237-f001:**
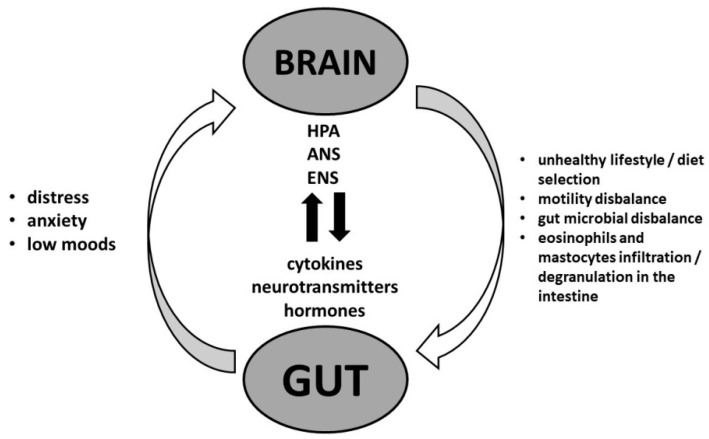
The interaction between metabolic syndrome and functional dyspepsia involving the gut–brain axis; HPA—the hypothalamic–pituitary–adrenal axis; ANS—the autonomic nervous system; ENS—the enteric nervous system.

**Figure 2 ijerph-21-00237-f002:**
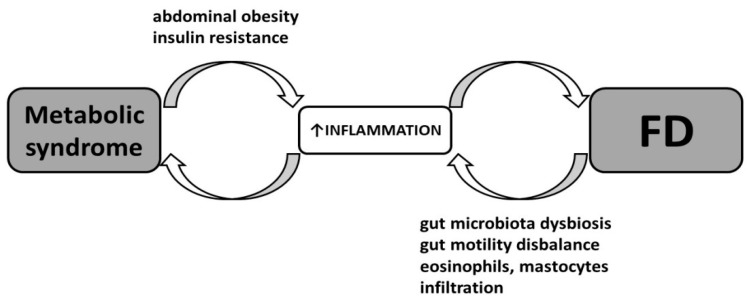
Crosstalk between obesity/metabolic syndrome and functional dyspepsia (FD).

**Figure 3 ijerph-21-00237-f003:**
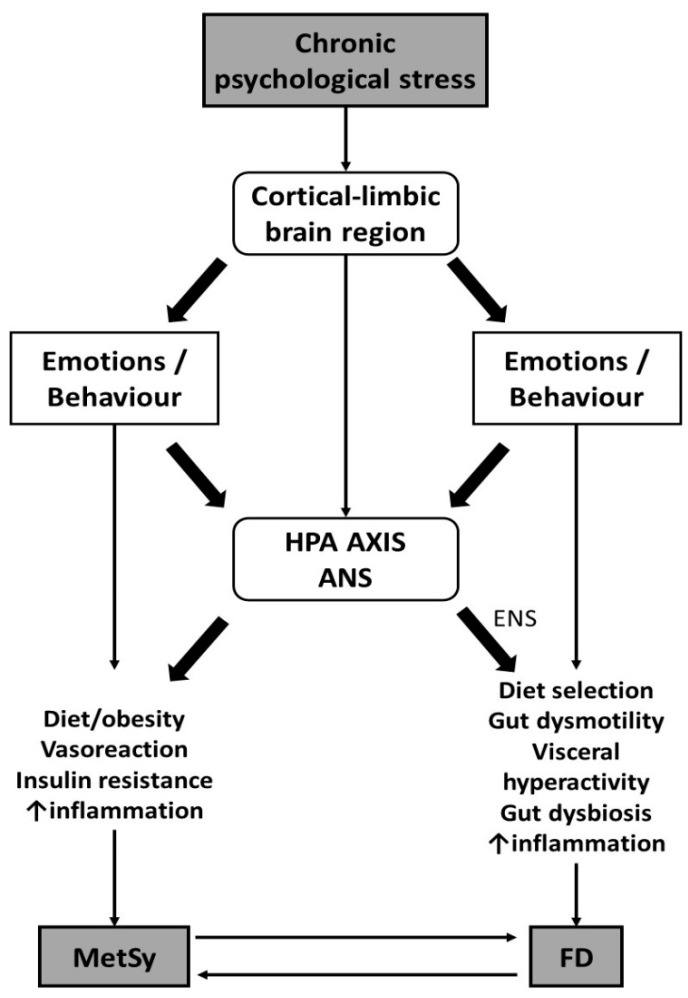
Chronic psychological stress as the common denominator of metabolic syndrome and functional dyspepsia.

## Data Availability

Not applicable.

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
