# Peer review of "The Association between Functional Dyspepsia and Metabolic Syndrome—The State of the Art"

_ijerph, 2024, doi:10.3390/ijerph21020237_

Round 1
Reviewer 1 Report
Comments and Suggestions for Authors
It is a good review paper on the relationship between FD and MetS. I think it will be a better paper if it supplements the following.
1. Recently, the importance of complementary and alternative medicine has increased and interest is also increasing. I hope that the relevance of traditional medicine, such as whether there is any traditional medicine that can be treated at the same time as FD and MetS, should be supplemented.
2. Ion channels are also one of the important factors for FD and MetS. I hope to supplement the relevance of ion channels.
Reviewer 2 Report
Comments and Suggestions for Authors
Dear authors and editor,
The manuscript titled "An association between functional dyspepsia and metabolic syndrome – a review" aimed to relate on the one hand the impact of psycholoigic stress with the increased prevalence of functional dyspepsia and metabolic syndrome. On the other hand, the authors aimed to investigate the relationship between functional dyspepsia and metabolic syndrome.
There are many minor and mayor issues I'd like the authors resolve.
Abstract
1-Change the keywords. Delete the words "functional gastrointestinal disorders", "microbiome", and "eating habits" . Not found in the MeSH (Medical Subject Headings).Change to "Gastrointestinal Diseases", "Gastrointestinal Microbiome", Stress, Psychological and "Feeding Behavior"
2-The authors do not make the study design clear in the title. It may be a narrative review or a systematic review or a scoping review or a state of the art.
3-Reformulate the title according to the subject matter.
Introduction
3-Adequate: The most important concepts of the subject to be developed are identified.The authors have done a great job in describing the findings.
Methodology and Discussion
4- If the authors intend to carry out a review, this section should be included in order to assess the quality of the review, as well as to be able to be reproduced by other authors.
Future prospects and conclusions
5-They would be necessary for both the review and the state of the art. This section does not appear as such in the manuscript.
The problem with the manuscript is that it is not clearly defined what type of design the authors want to use. Everything indicates that they are dealing with a state of the art.
Reviewer 3 Report
Comments and Suggestions for Authors
The article was an extensive review of dyspepsia and metabolic syndrome. It was an extensive look into connections between the two. I'll admit I was a tad disappointed at the fairly short section concerning psychosocial factors and development of issues - and honestly that's my one suggestion. That's a topic I know more about to review. There is quite a bit of data concerning stress and ulcers e.g., so it seems there should be more overlap than what is mentioned. My only other issue is with the English - editing is required. Examples include line 12 of the abstract "...it is not found an explanation in ..." ; line 135 "...in the recent decays." Maybe decades? Anyway those are two examples.
Comments on the Quality of English LanguageAs I noted above:
My only other issue is with the English - editing is required. Examples include line 12 of the abstract "...it is not found an explanation in ..." ; line 135 "...in the recent decays." Maybe decades? Anyway those are two examples.
Round 2
Reviewer 1 Report
Comments and Suggestions for Authors
It is well revised.
Reviewer 2 Report
Comments and Suggestions for Authors
Dear authors and editor,
I thank the authors for their efforts to improve the comprehensibility of the manuscript as well as to correct methodological flaws.
I consider that it has been correctly recondensed and the design they have used to develop the topic has been taken into account.
Thank you for the opportunity to read your research.
Best regards.